# TEAD Inhibitors Sensitize KRAS^G12C^ Inhibitors via Dual Cell Cycle Arrest in KRAS^G12C^-Mutant NSCLC

**DOI:** 10.3390/ph16040553

**Published:** 2023-04-06

**Authors:** Salvina Laura Tammaccaro, Philippe Prigent, Jean-Christophe Le Bail, Odette Dos-Santos, Laurent Dassencourt, Myriam Eskandar, Armelle Buzy, Olivier Venier, Jean-Claude Guillemot, Yaligara Veeranagouda, Michel Didier, Emmanuel Spanakis, Tokuwa Kanno, Matteo Cesaroni, Stephane Mathieu, Luc Canard, Alhassan Casse, Fanny Windenberger, Loreley Calvet, Laurence Noblet, Sukhvinder Sidhu, Laurent Debussche, Jurgen Moll, Iris Valtingojer

**Affiliations:** 1Oncology, Sanofi R&D, 94400 Vitry-sur-Seine, France; 2Bio Structure and Biophysics, Sanofi R&D, 94400 Vitry-sur-Seine, France; 3Small Molecules Medicinal Chemistry, Sanofi R&D, 91380 Chilly-Mazarin, France; 4Genomics and Proteomics, Translational Sciences, Sanofi R&D, 91380 Chilly-Mazarin, France; 5Precision Oncology, Sanofi R&D, 94400 Vitry-sur-Seine, France; 6Molecular & Digital Histopathology, Sanofi R&D, 94400 Vitry-sur-Seine, France; 7Non-Clinical Efficacy and Safety, Sanofi R&D, 94400 Vitry-sur-Seine, France; 8Pharmacology, Sanofi R&D, 94400 Vitry-sur-Seine, France

**Keywords:** YAP1-TEAD, resistance, KRAS^G12C^, NSCLC, cell cycle arrest

## Abstract

KRAS^G12C^ is one of the most common mutations detected in non-small cell lung cancer (NSCLC) patients, and it is a marker of poor prognosis. The first FDA-approved KRAS^G12C^ inhibitors, sotorasib and adagrasib, have been an enormous breakthrough for patients with KRAS^G12C^ mutant NSCLC; however, resistance to therapy is emerging. The transcriptional coactivators YAP1/TAZ and the family of transcription factors TEAD1-4 are the downstream effectors of the Hippo pathway and regulate essential cellular processes such as cell proliferation and cell survival. YAP1/TAZ-TEAD activity has further been implicated as a mechanism of resistance to targeted therapies. Here, we investigate the effect of combining TEAD inhibitors with KRAS^G12C^ inhibitors in KRAS^G12C^ mutant NSCLC tumor models. We show that TEAD inhibitors, while being inactive as single agents in KRAS^G12C^-driven NSCLC cells, enhance KRAS^G12C^ inhibitor-mediated anti-tumor efficacy in vitro and in vivo. Mechanistically, the dual inhibition of KRAS^G12C^ and TEAD results in the downregulation of MYC and E2F signatures and in the alteration of the G2/M checkpoint, converging in an increase in G1 and a decrease in G2/M cell cycle phases. Our data suggest that the co-inhibition of KRAS^G12C^ and TEAD leads to a specific dual cell cycle arrest in KRAS^G12C^ NSCLC cells.

## 1. Introduction

The Kirsten rat sarcoma viral oncogene homolog G12C isoform (KRAS) belongs to the RAS protein family of small guanosine–triphosphate hydrolases (GTPases) [1]. The KRAS^G12C^ mutation represents the most prevalent in lung cancer (14%) and is associated with carcinogen exposure, such as cigarette smoke [2]. The discovery of a druggable pocket on the KRAS protein and the recent development of covalent KRAS^G12C^ inhibitors with clinical efficacy have been enormous breakthroughs for prolonging the survival of patients with advanced NSCLC [3,4,5]. Sotorasib was the first approved KRAS^G12C^ inhibitor (KRAS^G12C^i) by the FDA as a monotherapy of second-line treatment for patients with advanced KRAS^G12C^ mutated NSCLC [4]. In parallel, adagrasib also showed a response in KRAS^G12C^ mutant NSCLC patients previously treated with chemotherapy and anti-PD1/L1 [6]. Despite these new molecules’ benefits, response rates remain below those obtained for other targeted therapies and range around 45% [2,7,8]. In addition, mechanisms of acquired resistance are emerging, including mutations in the KRAS protein, which prevent compound binding or activation of molecular pathways bypassing KRAS signaling [9,10,11].

The Hippo-YAP1 pathway is emerging as an effective bypass mechanism to mitogen-activated protein kinase (MAPK) pathway inhibition [12,13]. Hippo-YAP1 acts via the activation of transcriptional coactivator proteins, transcriptional coactivators yes-associated protein 1/transcriptional activator with PDZ-binding motif (YAP1 and TAZ), which interact with transcription factors from the TEAD-family (transcriptional enhanced associate domain 1-4; DNA binding transcription factors) and thereby regulate the transcription of genes required for cell survival and cell proliferation [14].

Several preclinical studies highlight the role of YAP1/TAZ-TEAD activation as a bypass mechanism to mutant KRAS inactivation. In a KRAS^G12D^-dependent mouse pancreatic ductal adenocarcinoma (PDAC) model, YAP1 gene amplification bypasses KRAS inactivation, leading to cancer progression. In a KRAS^G12D^-dependent mouse NSCLC model, the knockdown of YAP1 delays tumor regrowth [15]. In parallel, the loss of the wild-type KRAS allele can promote YAP1 activation, as wt-KRAS favors the cytoplasmic localization of YAP1 [16].

The crosstalk between KRAS and YAP1 pathways is partly mediated via the src tyrosine kinases (SFKs) family. These kinases modulate KRAS phosphorylation by either signaling on LATS1/2 or by directly acting on YAP1 phosphorylation [17]. Additional interaction nodes between YAP1 and KRAS pathways that have been suggested include (i) the convergence of both pathways on the activation of the induction of epithelial–mesenchymal transition (EMT) via the transcription factor fos proto-oncogene (FOS), (ii) the KRAS-mediated stabilization of YAP1 via the downregulation of SOCS-box proteins and (iii) the role of YAP1 as an oncogenic KRAS effector in TRP53 mutant pancreas tumor [15,18,19].

Despite the described crosstalks, the direct effects of combining a KRAS^G12C^i with a TEADi have yet to be investigated. Here, we explore the combination of KRAS^G12C^ and YAP1-TEAD inhibition on KRAS^G12C^-NSCLC cell lines. We performed drug combination studies using KRAS^G12C^i adagrasib (MRTX849) or sotorasib (AMG510) and pan-TEADi K-975 or Sanofi compound A (Cpd A) for the in vivo combination study. We evaluated the effect of this dual inhibition on tumor cell growth and biomarker modulation and characterized the effects of the combination at the transcriptomic, proteomic, and phospho-proteomic levels.

## 2. Results

### 2.1. TEAD Inhibition Potentiates A KRAS^G12C^ Inhibition in NSCLC Cell Lines

We selected nine KRAS^G12C^ NSCLC cell lines and investigated the effect of K-975 and adagrasib on the phenotype of each of the selected cell lines alone or in combination. We observed that treatment with K-975 alone had no or only weak effects on the growth of the selected cell lines. At the same time, adagrasib inhibited cell growth in seven out of nine KRAS^G12C^ mutant cell lines with a range of half-maximal inhibition (IC_50_) (Figure 1). We then monitored the effect of the dual inhibition by combining ten doses of adagrasib (from 0.3 to 10,000 nM) with two doses of K-975 (100 and 1000 nM) in a cell growth assay at 144 h.

Overall, the results from this experiment allowed us to detect a better response when adagrasib was combined with K-975 for all the cell lines tested except for two (SW1573 and NCI-H23). Concerning NCI-H2030, HOP 62, NCI-H1792, and NCI-H358, we showed a beneficial effect for both concentrations of K-975 tested (100 and 1000 nM). In comparison, for NCI-H1373 and LU99A, at least one concentration of K-975 (either 100 or 1000 nM) significantly improved the IC_50_ of adagrasib (Figure 1).

We next performed a cell growth inhibition study on HOP 62 and NCI-H2030 cell lines using a more extensive range of concentrations for both inhibitors. In this confirmatory study, we combined the compounds with fourteen doses of K-975 and eighteen doses of adagrasib to assess growth inhibition at 144 h. This analysis aimed to verify the presence of synergy or additivity. However, since K-975 is an inactive compound regarding cell growth inhibition in the cell lines we tested, we could not statistically calculate synergy or additivity, they both require the activity of each compound as a single agent.

We therefore explored the statistical effect of what is called potentiation or potentialization. The calculation of potentiation allows us to statistically calculate the effect of a compound, which is inactive (in this case K-975), on the growth inhibitory effect of another compound (in this case adagrasib) in a given cell line. For a statistically correct analysis, it is required that the so-called inactive compound is used at doses at which it is really inactive, i.e., with a percentage of cell growth inhibition below 20% in the combination study.

When performing this type of analysis with K-975 in such dose ranges in combination with adagrasib, we can determine a reduction in the IC_50_ value of adagrasib by 4–6 times in NCI-H2030 (Figure 2A, values in the box) and by three times in HOP 62HOP-62 (Figure 2B, values in the box) cell lines, statistically confirming the potentiation effect of the by-itself inactive K-975 on adagrasib.

### 2.2. TEADi Combined with KRAS^G12C^i Impacts Genes Implicated in Cell Cycle Regulation at G1/S Transition and G2/M Checkpoint

Before investigating the molecular mechanisms implicated in the observed combination effects, we determined the degree of target inhibition obtained with K-975 on TEAD and with adagrasib on KRAS-G12C in the single and combination study treatments. Our treatment conditions studied were: (i) DMSO-vehicle control, (ii) adagrasib alone (30 nM), (iii) K-975 alone (100 nM), and (iv) combination of the inhibitors at the selected single-treatment doses (COMBO). All samples were collected at 24 h post-dosing, and RNA sequencing followed by gene set enrichment analysis (GSEA), liquid chromatography mass spectrometry analysis (LC/MS), and target occupancy ratio (TOR) studies were run (Figure 3A).

We focused our study on TEAD1 and TEAD4, as they were the most abundant protein isoforms by LC/MS in the NCI-H2030 cell line (TEAD1 = 1.5 fmol, TEAD4 = 0.7 fmol, data not shown). TOR demonstrated specific binding of compounds to their target protein; 90% of TEAD1 and 50% of TEAD4 were occupied by K-975, while 80% of KRAS^G12C^ was occupied by adagrasib (Appendix A). Data were comparable for each target in the single and the combination treatments.

Next, we characterized target pathway-specific biomarker modulation for K-975 and adagrasib. We introduced a gene-reporter element carrying the TEAD-responsive promoter and a luciferase gene reporter (TEAD-Luc) to NCI-H2030 and HOP 62 cell lines. We used this gene reporter as a biomarker for YAP1-TEAD activity. As expected, we detected potent inhibition of TEAD-Luc by K-975 (IC_50_ of 10–20 nM) in NCI-H2030 (TEAD-Luc) and HOP 62 (TEAD-Luc) cells and observed no effect for adagrasib (Appendix A). Adagrasib, on the other hand, potently inhibited phosphorylation of the KRAS biomarker ERK1/2 at Thr202/Tyr204 and Thr185/Tyr187 (IC_50_ < 10 nM), while K-975 was only weakly active in this assay (IC_50_ = 30,000 nM) (Appendix A).

We then performed a transcriptomic analysis on the different samples and looked at the overall transcription change in the combination versus the single-agent treatment. We focused on transcription signatures specifically modulated in the combination studies and grouped these signatures by signaling pathways with the help of GSEA analysis [20]. We identified a potent inhibition of signatures representative of E2F_TARGETS, G2M_checkpoint, and MYC_TARGETS_V2 (Figure 3B) in the combination setting, considering a false discovery rate (FDR) *q* value < 0.1. To see how the changes observed on the transcriptome level would translate to the protein level, we performed global proteomic analysis by mass spectrometry.

These studies identified 10 proteins modulated by K-975 and 19 proteins modulated by adagrasib in the respective single-agent treatment. In comparison, 62 out of a total of 5000 proteins were modulated in the combination treatment, considering fold change (FC) ≥2 and *p* value < 0.01 (Figure 3C). Among the proteins modulated in the combination studies, the most substantial modulation was observed for proteins involved in cell cycle modulation. Gene Ontology (GO) and pathway map analysis on the proteomic data showed that the modulated proteins are implicated in pathways representative of cell division and mitosis, specifically anaphase promoting the complex/cyclosome (APC/C) and metaphase checkpoint. These signaling pathways are downregulated in the combination treatment. We found some biomarkers modulated such as cell division cycle protein 20 homolog (CDC20), serine/threonine kinase polo-like kinase 1 (PLK1), diaphanous related formin 3 (DIAPH3) and BUB1 mitotic checkpoint serine/threonine kinase B (BUB1B; Figure 3D).

### 2.3. KRAS^G12C^ and TEAD Directly Downregulate Genes Involved in the S Phase Transition and the Assembly of the Mitotic Spindle

We further validated results in transcriptomic and proteomic experiments with YAP/TAZ-TEAD ChIP-Seq from the Zanconato et al. study, where 3089 TEAD responsive elements on enhancer and promoter regions have been found [21]. We identified the overlap between these 3089 elements and the transcripts/proteins modulated in the combination treatment and obtained a short list of 33 genes specifically regulated by the combination. Significance thresholds were set at *p* value < 0.01 for RNA and protein. The FC thresholds were set to 1.5 for RNA and 2 for proteins.

We detected 14 genes that were presenting TEAD-responsive elements (called direct targets) and 19 genes without TEAD-responsive elements (named here indirect targets; Figure 4A). We found that the combination modulates the expression of genes involved in S phase regulation and genes involved in the regulation of the mitotic spindle assembly (Appendix A). Using this information, we probed the most strongly modulated biomarkers implicated in the pathway of the correct kinetochore attachment to the mitotic spindle by Western blot (via capillary assay JESS). We confirmed the downregulation of candidate targets where the combination was more effective than the single-agent treatment (such as BUB1, BUBR1, CDC20, PLK1) (Figure 4B and Appendix A). As internal control of TEADi activity, we used the cysteine-rich angiogenic inducer 61 (CYR61) biomarker, which shows modulation alone or combined with adagrasib (Figure 4B and Appendix A).

### 2.4. KRAS^G12C^ and TEAD Have a Dual Impact on Cell Cycle Progression

We proceeded to investigate if combination treatment impacts cell cycle progression. NCI-H2030-CC and NCI-H1792-CC cell lines, stably expressing the fluorescent ubiquitinated cell cycle indicator (FUCCI), were created to detect different stages of cell cycle arrest. In a time-course experiment up to 72 h, we observed that every single agent led to a slight arrest in the G1 phase post-treatment (Figure 5A). The combination of adagrasib and K-975 showed more robust G1 phase arrest than single agents (Figure 5A). The difference between the combination vs. adagrasib alone was statistically significant (*p* value < 0.0001 at 24 h for NCI-H1792-CC and 48 h for NCI-H2030-CC). The combination showed a more prolonged G1 phase than the DMSO control (Figure 5B).

We decided to explore in parallel if the combination treatment gave the same arrest as Aurora A/B Kinase inhibitor (SML0268) or PLK1 inhibitor (RO-3280), two compounds modulating mitotic checkpoint proteins that are known to induce mitotic catastrophe and polyploidy. Even though Aurora B kinase and PLK1 proteins were themselves modulated by KRAS^G12C^i and TEADi combination, the cell cycle effect induced by the combination treatment differed from the cell cycle effect induced by either SML0268 or RO-3280 (Figure 5A).

The KRAS^G12C^i and TEADi combination induced dual cell cycle arrest, which was also confirmed by the analysis of cellular DNA content with fluorescence-activated single-cell sorting (FACS) and by cytoskeleton density and DNA ploidy analysis via immunofluorescence (IF). FACS analysis confirmed a more extended G1 phase at 48 h (*p* value < 0.0002 for the combination vs. adagrasib alone) (Figure 5C).

We also obtained a shortened G2/M phase with the combination (*p* value 0.0027 for the combination vs. adagrasib alone), which suggested a combination-specific dual action on the cell cycle (Figure 5C). We did not observe any anomaly in the DNA content by IF. However, we saw a 1.5–2 fold increase in beta-tubulin density in the COMBO compared to all the other conditions (*p* value < 0.0001; Appendix A).

Finally, we assessed phosphoproteomic changes after combination treatment with the Post Translational Modification (PTM) scan assay, from Cell Signaling Technology (CST), in the following conditions: DMSO, adagrasib, and sotorasib at 30 nM, alone or in combination with 100 nM of K-975 (Appendix A). The immunoprecipitation with phosphotyrosine (pY) assay revealed seven phosphorylated targets specific to the combinations. Beta-tubulin showed the strongest dephosphorylation that was combination specific. This dephosphorylation was seen with both KRAS inhibitors tested (Appendix A). By Immobilized Metal Affinity Chromatography (IMAC) assay, we detected 25 phosphorylated peptides in the combinations. Most were implicated in ubiquitination and sumoylation processes involved in the transition G2/M or in the correct kinetochore assembly (Appendix A). These data confirmed the impact of the combination on the G1 and G2/M phases previously observed.

### 2.5. Sanofi Proprietary TEADi Ameliorates Adagrasib Activity by Reducing Tumor Cell Growth in NSCLC Cell Line KRAS^G12C^ NCI-H358

We decided to confirm our results in an in vivo study. To this end, we first evaluated the tumor formation of NCI-H2030, HOP 62, H1792, and NCI-H358 cell lines implanted as subcutaneous models in vivo. Among these four cell lines tested, only the NCI-H358 cell line led to rapid tumor formation in vivo and was selected as the xenograft model of choice for in vivo evaluation of the KRAS^G12C^i-TEADi combination. We then selected the best compounds to use for this study. While K-975 is an efficacious tool compound for in vitro studies, this molecule is suboptimal for in vivo combination studies due to its strong CYP-induction potential. We therefore decided to use the Sanofi TEADi here, called Cpd A, as a TEADi for this in vivo drug combination study. Cpd A also binds to the central lipid pocket of TEAD(1-4) proteins such as K-975 and exerts equal potency in relevant cellular assays (data not shown). However, it does not affect CYP induction or inhibition and is better suited for in vivo combination experiments.

After a first in vitro evaluation in which we confirmed that the combined effect of Cpd A with adagrasib compares to what we previously observed for K-975 (Appendix A), we tested this compound in vivo alone and in combination with adagrasib. Similar to our cellular results, while Cpd A was inactive when tested as a single agent at 25 mg/kg QD, and adagrasib tested at 10 mg/kg QD was marginally active, the combination of Cpd A with adagrasib demonstrated significantly superior activity compared to adagrasib alone (*p* value < 0.0001; Figure 6). We thus confirmed the beneficial effect of a TEAD inhibitor in vivo on adagrasib-mediated growth inhibition.

## 3. Discussion

Here, we report an integrative analysis of the effects of combining a KRAS^G12C^ inhibitor with a YAP1/TAZ-TEAD inhibitor in KRAS^G12C^ mutant NSCLC cell lines. We show that adding a TEADi potentiates the effect of adagrasib in most cell lines tested. Mechanistically, we demonstrate that the effect of the combination leads to a dual cell cycle arrest characterized by blockage in the G1 and G2/M phases. Previous studies have reported that YAP1-TEAD complexes regulate genes implicated in the cell cycle [21], including some of the genes we identify in the current study. For example, HELLS and ATAD2 are coactivators that favor the expression of genes implicated in G1/S transition and S phase progression, which we see inhibited in our combination treatment. They drive cell proliferation through the activation of MYC in the case of ATAD2 and E2F in the case of HELLS [22]. MYC and E2F were the main transcription signature programs downregulated in our KRAS^G12C^/TEADi combination.

The other prominent signature modulated by our combination was the G2/M checkpoint. Some data are available on YAP1 activation in the context of resistance to EGFR inhibitor treatment. In EGFR mutant NSCLC cells, the YAP1/FOXM1 axis mediates resistance to EGFR inhibitors by increasing the expression of spindle assembly checkpoint (SAC) proteins [23]. SAC components play an essential role in the correct attachment of microtubules (MT) of the mitotic spindle to the kinetochore (KT) by inhibiting the formation of the APC/C-CDC20 complex [24]. APC/C is an E3 ubiquitin ligase necessary for mitotic exit [25,26,27]. SAC proteins inhibit the formation of the APC/C-CDC20 complex when one KT is unattached or is incorrectly attached [25].

Our studies show that downregulation of the KRAS-YAP1-TEAD pathway also inhibits the expression of essential modulators such as Aurora B and PLK1 kinases at transcript and protein levels. These kinases are known components of SAC that regulate the protein levels of BUB1, BUBR1, and CDC20, which are all implicated in regulating APC/C-CDC20 complex formation. We validated the protein decrease in some of these SAC components in response to the combination treatment.

Based on the literature data and our findings, we propose a model where the combination of KRAS^G12C^ and TEAD inhibition converges in the downregulation of genes required for cell cycle progression and kinetochore attachment mediated by transcription factors (TFs) such as FOXM1 (Figure 7). In this model, the combination may, on the one hand, inhibit the transcription of HELLS and ATAD2 via FOXM1, leading to an arrest in G1 [28]. On the other hand, potential FOXM1 inhibition may decrease the expression of SAC components, inhibiting the formation of the APC/C complex and leading to reduced entry into the mitotic phase.

FOXM1 can modulate the expression of several genes, some of which are implicated in specific checkpoints of cell cycle progression [29,30]. Cell cycle analysis showed a reduction of the S/G2/M phases and no changes in late mitosis, a phenotype different from the pure inhibition of SAC components. This observation can be explained by the fact that KRAS^G12C^i and TEADi combination inhibits the SAC, E2F, and MYC transcription programs, resulting in a dual cell cycle effect. Whether this dual cell cycle inhibition event occurs in parallel or whether one precedes the other will need to be investigated. Furthermore, additional evaluations of other cell lines and a detailed genomic study will be required to understand better this phenomenon suggested in the literature [31,32] and whether the effect on cell cycle inhibition would be comparable.

## 4. Materials and Methods

### 4.1. Cell Lines Culture Conditions

Cell lines were maintained at Sanofi and grown at 37 °C under 5% CO_2_, except for SW1573, which was at 0% CO_2_. A short tandem repeat assay authenticated all cell lines at the Microsynth AG (Balgach, Switzerland). PCR using the Venor^®^GeM kit excluded mycoplasma infection (Biovalley, MB minerva biolabs, Skillman, NJ, USA). HOP 62 (#CRL-11350) RRID: CVCL_1285, NCI-H2030 (#CRL-5914) RRID: CVCL_1517, NCI-H1792 (#CRL-5895) RRID: CVCL_1495, NCI-H23 (#CRL-5800) RRID: CVCL_1547, NCI-H358 (#CRL-5807) RRID: CVCL_1559, NCI-H1373 (#CRL-5866) RRID: CVCL_1465, NCI-H2122 (#CRL-5985) RRID: CVCL_1531, SW1573 (#CRL-2170) RRID: CVCL_1720 were purchased from the American Type Culture Collection (ATCC), while LU99A was purchased from the Japanese Collection of Research Bioresources Cell Bank (JCRB0044) and cultured according to supplier’s recommendation. NCI-H2030 and HOP 62 were stably transduced either with the Incucyte^®^ Cell Cycle Green/Orange Lentivirus reagent (#4809, Sartorius, Göttingen, Germany) or with TEAD Luciferase lentiviral construct (FlashTherapeutics, Toulouse, France) containing a TEAD-responsive synthetic element/region (8xGTIIC region followed by a minimal chicken TNNT2 promoter, driving luciferase expression). All transduced cells were cultured according to the supplier’s recommendations, plus the puromycin or neomycin (#10781691 and #G8168, Sigma, St Louis, MO, USA) to maintain the selected population.

### 4.2. Luciferase Assay

NCI-H2030 and HOP 62 cells were seeded at 8 × 10^4^ cells/well density in 96-well plates in a complete culture medium and incubated at 37 °C in 5% CO_2_ for 24 h. Initially dissolved in DMSO, compounds were added to the culture medium at different dilutions with a final amount of 0.1% DMSO. After 24 h of incubation, luciferase activity was detected by a luminescence plate reader (Spark, TECAN Männedorf, canton of Zürich, Switzerland) using Bright-Glo™ Luciferase Assay System (according to the manufacturer’s instruction).

### 4.3. Compounds Used

We used the TEADi K-975 [33], the KRAS^G12C^ inhibitors adagrasib [34] and sotorasib [35], the inhibitors for PLK1 (RO-3280) and Aurora B Kinase (SML-0268), and a proprietary compound from Sanofi (Cpd A).

### 4.4. Compound Combination Assay for A Confirmatory Study

Cell lines were seeded in 384-well plates in a complete culture medium. Cell lines were seeded at 150 cells/well for HOP 62 and 125 cells/well for NCI-H2030. After 24 h at 37 °C and 5% CO_2_, compounds were added at different dilutions to have a final percentage of 0.2% DMSO in the culture medium. Single compound treatment was performed using a range of fourteen concentrations for adagrasib from 10,000 to 0.006 nM and eighteen concentrations for K-975 from 10,000 to 0.0001 nM. For combination treatments, the same concentrations and doses were used for each inhibitor, leading to 14 × 18 combination points in this large compound combination assay. Cell viability was read after 144 h of treatment using the CellTiter-Glo^®^ Luminescent Cell Viability Assay (Promega, Madison, WI, USA) reagent with an incubation time of 1 h at RT. Luminescence was detected using the TECAN SPARK plate readers. To characterize the interactions between adagrasib and K-975 on NCI-H2030 and HOP 62 cells, the IC50 of adagrasib alone and with K-975 were estimated from ranges of adagrasib between 0.02 and 10,000 nM alone and mixed with eleven constant concentrations of K-975 between 0.5 and 3333 nM. The ratio R = IC_50adagrasib with K-975_/IC_50adagrasib alone_ was calculated with its confidence interval with SAS^®^ version 9.4. If R is significantly lower than 1 (i.e., the lower bound of its confidence interval is lower than 1), the IC_50_ of K-975 with adagrasib is significantly lower than the IC_50_ of adagrasib alone. Moreover, if K-975 is nonactive, a potentiation can be concluded. Three independent experiments were conducted to obtain six replicates of the wells with single compounds and triplicates of the wells with combinations.

### 4.5. Compound Combination Assay

Cell lines were seeded in 96-well plates in complete culture medium and cultured at densities allowing for exponential growth: 1250 cells/well (HOP 62), 400 cells/well (NCI-H2030), 500 cells/well (NCI-H1792), 5000 cells/well (NCI-H23), 2500 cells/well (NCI-H353, SW1573), 600 cells/well (NCI-H1373), 1000 cells/well (NCI-H2122), and 1200 cells/well (LU99-A). After 24 h, compounds were added in different dilutions with a final percentage of 0.2% DMSO. Single-agent treatments were performed with a range of ten concentrations per compound (0.3 to 10,000 nM) and a dilution factor of approx. 3x between each of the concentrations. In the combination study, adagrasib was tested at the same range of ten concentrations used in the single-agent studies, while K-975 was added at two fixed doses (100 and 1000 nM). Cell viability was determined after 144 h of compound treatment using the CellTiter-Glo^®^ Luminescent Cell Viability Assay (Promega, Madison, WI, USA) reagent with incubation of 1 h at room temperature (RT). Luminescence was detected using the TECAN SPARK plate readers. IC_50_ values were calculated using Excel Fit on three independent experiments.

### 4.6. Phosphorylated/Total ERK1/2 Detection

HOP 62 and NCI-H2030 cells were seeded at 6000 cells/well in 96-well plates in complete culture medium at 37 °C at 5% CO_2_. After 24 h, compounds were added at different dilutions to obtain a final percentage of 0.1% DMSO. Single compounds were tested using a range of ten concentrations from 0.0003 μM to 10,000 nM for both inhibitors, adagrasib, and K-975. After 5 and 24 h of compound treatment, the culture medium was removed, and the cells were lysed with the cell extraction buffer (#FNN0011, Invitrogen, Waltham, MA, USA), complemented with Anti-Protease (#04963124001, Roche, Basel, Switzerland) and 1 mM PMSF (#93482, Sigma). Protein extracts were processed according to MSD^®^, Rockville, Maryland, MULTI-SPOT Assay System (#K15107D) protocol. IC_50_ values were calculated using Excel Fit on three independent experiments.

### 4.7. RNA Extraction, HiCAR-Seq Library Preparation, Sequencing, Analysis

RNA was isolated from frozen cell pellets using RNEASY Plus Mini Kit (#74136, Qiagen, Venlo, Netherlands). Then, 50 ng of total RNA was used for HiCARSeq library preparation. HiCARSeq library was prepared as described previously [36]. HiCARSeq libraries were sequenced along with other samples using NVSEQ 6000 S1 Rgt Kit v1.5 (100 cyc) and NovaSeq 6000 Sequencing System (Illumina, San Diego, CA, USA). NovaSeq 6000 output files were converted into FASTQ files using bcl2fastq and used in downstream analysis. HiCARSeq transcriptome FASTQ files were analyzed on Array studio V10.0 (Omicsoft, Qiagen). Following raw-read QC, the last few bases were trimmed and mapped to the reference genome Human.GRCh38. The read count data were generated using GeneModel RefGene20200204. The resulting data were normalized by the DESeq package, transformed to a log2 value, and used for pathway prediction and one-way ANOVA analyses. For pathway enrichment, GSEA was performed using the Hallmark gene set of Molecular Signatures Database v7.4 [20]. To estimate the TEAD-dependent transcription rate and to evaluate TEAD inhibitors’ pharmacodynamics, we used a transcriptional signature of TEAD activity and a scoring method described elsewhere [37].

### 4.8. Sample Preparation and Liquid Chromatography Mass Spectrometry Analysis (LC/MS)

NCI-H2030 cells were resuspended in cell lysis buffer (#FNN0011, Invitrogen) supplemented with protease and phosphatase inhibitor cocktail (#78446, Thermo Fisher Scientific, Waltham, MA, USA). The mixture was then stirred for 10 min at 70 °C, followed by centrifugation at 10,000 rpm for 10 min. Protein concentration was measured by BCA analysis, and 30 μg of total proteins from cell lysates was diluted in Laemmli buffer and loaded on a stacking gel. The top of the front gel was excised, reduced with DTT, alkylated with iodoacetamide, and in-gel digested with trypsin. Peptides were extracted with 50 mM ammonium bicarbonate and 50% acetonitrile in 0.2% formic acid, dried by evaporation in a speed-vac concentrator, and resuspended in 60 μL of 0.2% formic acid. Samples were spiked with six heavy isotopic AQUA standards peptides at 10 fmol each (Thermo Fisher Scientific). All samples were separated into two aliquots, one analyzed in Data Dependent Acquisition (DDA) mode LC/MS analysis for global proteome differential analysis and the other one subjected to targeted LC/MS analysis for TEAD1, TEAD2, TEAD3, and TEAD4 expression level determination and target occupancy ratio (TOR) measurement. Analyses were performed using a nano-ACQUITY Ultra-Performance LC system (Waters, Milford, MA, USA) coupled to an Orbitrap Fusion Tribrid mass spectrometer (Thermo Fisher Scientific). LC separation was performed with a trapping column (nano-Acquity Symmetry C18, 100 Å, 5 μm, 180 μm × 20 mm) at 15 μL/min flow rate and an analytical column (nano-Acquity BEH C18, 130 Å, 1.7 μm, 75 μm × 250 mm) directly coupled to the ion source. The mobile phases for LC separation were 0.2% (*v*/*v*) formic acid in LC-MS grade water (solvent A) and 0.2% (*v*/*v*) formic acid in acetonitrile (solvent B). Peptides were separated at a 300 nl/min constant flow rate with a linear gradient of 5–85% solvent B for 120 min for global proteome analysis and 35 min for targeted LC/MS. For DDA experiments, a full MS1 survey scan was acquired in the Orbitrap for *m*/*z* 325–1200. The resolution was set to 120 k at *m*/*z* 200. Fragmentation was performed in the HCD fragmentation cell (collision energy at 26%), isolating precursor ions in the quadrupole. Target ions previously selected for fragmentation were dynamically excluded for 50 s with a relative mass window of ±10 ppm. TOR experiments were performed as recently described [37].

### 4.9. Data Processing for Proteomics Analysis and Pathway Analysis

DDA data were processed with the MaxQuant software (Ver. 1.6.17.0, Max-Planck Institute of Biochemistry, Department of Proteomics and Signal Transduction, Munich, Germany). Database searching was performed against the human FASTA database downloaded from UniProtKB/Swiss-Prot. Interrogation of the databank was based on the following criteria: precursor mass tolerance of 7 ppm, fragment ions mass tolerance of 0.6 Da, and 2 maximum missed cleavages with trypsin as the enzyme. Search parameters for post-translational modifications were variable modifications of *N*-acetylation on protein *N*-terminal residues, oxidation on methionine residues, and pyro-Glu modification on glutamine residues. The matching between runs was also checked. All the other parameters were the MaxQuant default parameters. Protein intensities were exported from the MaxQuant proteinGroups file. Missing values were replaced by the minimum value of each acquisition. LFQ intensities were transformed into their log2 values. Medians were calculated over the technical replicates. A two-tailed *t* test for each peptide was performed on the normalized medians to determine the statistical significance between vehicle and treated sample groups, assuming equal variance. The gene lists generated with *p* value < 0.01 and absolute fold change (FC) >2, after analysis of variance of the proteomic levels under various treatments, were submitted to Gene Ontology (GO) process and pathway enrichment analyses using Metacore (https://portal.genego.com/ (accessed on 18 November 2022)).

### 4.10. Statistical Analysis of Transcriptomic and Proteomic Levels

Differentially expressed genes were identified by one-way analysis of variance of quantile-normalized signals followed by a Scheffe test for multiple comparisons. The SPSS Statistics version 25 (IBM, Armonk, NY, USA) software was used for all statistical computations. Once this selection was made, the data in the different conditions for proteomic and transcriptomic datasets were compared with the data on the TEAD-responsive elements found in the YAP/TAZ/TEAD Chromatin Immunoprecipitation followed by Sequencing (ChIP-Seq) data [21] in Appendix A. Venn diagrams were drawn by Draw Venn Diagram (http://ugent.be (accessed on 16 August 2022)).

### 4.11. Protein Extraction and Quantification

NCI-H2030 cell line was seeded in 6-well plates in a complete medium. After 24 h, compounds were added at different dilutions to obtain a final percentage of 0.2% DMSO. After 24 h of treatment, cells were harvested using TrypLE Express (#12604013, Thermo Fisher Scientific) and washed in 1X PBS. Proteins were extracted by treating with RIPA buffer 1X (#89901, Thermo Fisher Scientific) supplemented by protease and phosphatase inhibitors 1X (#78446, Thermo Fisher Scientific) and DNase Basemuncher (#15417954, Thermo Fischer Scientific). Protein concentrations were determined using bicinchoninic acid method (BCA) (#23225, Thermo Fischer Scientific).

### 4.12. Capillary Western Immunoassay (Jess)

A total protein assay was conducted for data normalization and comparison on the Jess following the manufacturer’s instructions (ProteinSimple, San Jose, CA, USA) with a 12–230 kDa separation module (SM-W004, ProteinSimple). Target proteins were detected with antibodies from Abcam HAUS6 (ab173281), BUBR1 (ab215351), DIAPH3 (ab245660) and from Cell Signaling Technology (CST), Danvers, MA, USA, BUB1 (#5421) RRID:AB_10691315, CDC20 (#14866) RRID:AB_2715567, PLK1 (#4513) RRID:AB_2167409 and CYR61 (#14479) RRID:AB_2798492. The level was normalized to the total protein for each target protein studied, and then, the inhibition rate was calculated vs. DMSO. Three independent experiments were carried out.

### 4.13. Fluorescence-Activated Single-Cell Sorting (FACS)

The Guava Cell Cycle Reagent determines, by labeling cellular DNA with propidium iodide (PI), the percentage of cells in G0/G1, S, G2/M, and diploid phases based on DNA content. NCI-H2030 cell line was seeded in 6-well plates in a complete medium for 24 h. Then, compounds were added to cells. After 48 h incubation in compounds, cell cycle staining was realized according to Guava Cell Cycle Reagent Package Insert #4500-0220 (Luminex, Austin, TX, USA). Briefly, after harvesting cells with TrypLE Express (# 12604013, Thermo Fisher Scientific), washed in 1X PBS and counted (Vi-Cell XR, Beckman Coulter, Brea, CA, USA), cells were resuspended in polystyrene tube, and ice-cold 70% ethanol was slowly added. After overnight incubation at −20 °C, the equivalent volume of 150,000 cells was centrifuged at 450× *g* for 5 min and washed once with 1X PBS. On the cell pellet, 200 μL of Guava Cell Cycle reagent was added to each tube and incubated for 30 min at room temperature in the dark. Quantification of the cell cycle was realized with a flow cytometer from Luminex (Guava easyCyte 6HT-2L) and analyzed by CellCycle GuavaSoft 4.5 software. Recording excitation and emission wavelengths were 488 and 785 nm, respectively. Five thousand events were analyzed for each sample. At least three independent experiments were carried out. For data comparisons and statistical analyses, 2-way ANOVA and Tukey’s multiple comparisons test were used (GraphPad Prism v8). A *p* value < 0.05 was considered statistically significant.

### 4.14. Incucyte with Cell-by-Cell Detection

NCI-H2030-CC and NCI-H1792-CC cell lines were seeded at 1500 cells/well in 96-well plates in a complete medium for 24 h. The compounds were added at different dilutions to obtain a final percentage of 0.2% DMSO. Cells were then placed in INCUCYTE S5 (Sartorius, Göttingen, Germany). Photos were taken every 24 h at 10×. Data analysis was performed with a cell-by-cell adherent module with the IncuCyte software v. 2022A, as indicated by the supplier’s recommendations.

### 4.15. Immunofluorescence (IF)

NCI-H2030 cells were seeded at 5000 cells/well on glass coverslips placed into a 24-well plate. After 24 h, cells were then treated with different compounds. After 24 h of treatment, the media were gently aspirated, briefly rinsed with 1X PBS, and incubated with 600 μL of 4% formaldehyde (4 °C) per well for 20 min. The formaldehyde was aspirated, and the cells were rinsed with 1X PBS before a permeabilization step (Triton 0.1% 15 min at RT) and a saturation step with 1% of BSA, 20 min at RT. Immunofluorescence was performed using mouse anti-Tubulin-Alexa_488 antibody (#ab195887, Abcam, Boston, MA, USA) incubated at 4 °C overnight. Cells were rinsed and stained with DAPI before mounting the coverslips (0.17 mm) on a microscope slide. The observation was performed on the VS-120 Olympus scanner using the extended focus microscopy module to capture the cell volume even though the final image is 2D. A mixed model was performed on log-transformed tubulin density with treatment as fixed factor treatment and area into the well as random factor followed by a Dunnett’s test to adjust for the multiplicity of tests. Statistical analyses were performed using SAS^®^ version 9, USA: 5 770. A *p* value < 0.05 was considered statistically significant.

### 4.16. Post Translational Modification (PTM) Scan Analysis

NCI-H2030 cells were seeded in a complete medium at 200,000 cells in 500 cm^2^. After 72 h, compounds were added to the dishes. After 24 h incubation, cell lysis was realized according to the PTM scan urea protocol provided by CST. This experiment was realized three times for each condition. Samples extracts were sent to CST for PTM scan studies characterized by detecting phosphorylated tyrosine, serine, and threonine. Phosphorylated tyrosine (pY) was detected via immunoprecipitation, and phosphorylated serine and threonine were detected via Immobilized Metal Affinity Chromatography (IMAC). PTM scan data were processed by CST. Log2FC was calculated for each peptide expression, comparing each treatment condition against DMSO-vehicle control. Log2FC values were then normalized by subtracting the log2 ratio for each peptide from the median log2 ratio of all peptides within a treatment condition. Only peptides with normalized log2FC > 1 or < −1 were considered in downstream analysis. A single pooled sample (from three biological replicates) was processed per condition; no statistical tests were performed. PTM scan analysis was performed in R v.3.5.3.

### 4.17. In Vivo Efficacy Study

Female CB17/lcr-Prkdc^scid^/lcrIcoCrl mice (6–8 weeks old) were bred at Charles River (Les Oncins, Saint-Germain-Nuelles, France), housed in Sanofi AAALAC-accredited animal facilities, and were provided with irradiated food and filtered water ad libitum. All experiments were carried out following the French law and the European Directive 2010/63/EU for the protection of animals used for scientific purposes and with the approval of the ethics committee #21 (project number APAFIS#5644-2016061311593064.V1). In vivo efficacy study of Cpd A was performed in SCID mice inoculated subcutaneously into the right flank with 3 × 10^6^ NCI-H358 cells mixed with Matrigel. Mice bearing around 200 mm^3^ subcutaneous NCI-H358 tumors were randomly assigned to 5 groups of 8 mice/group and treated every day for 23 consecutive days with either vehicle, Cpd A 25 mg/kg, adagrasib 10 mg/kg or a combination of Cpd A and adagrasib. Tumor perpendicular diameters were measured twice per week with a caliper, and tumor volume (V) was calculated according to the following equation: V (mm^3^) = (d^2^ (mm^2^) × D (mm))/2, where d is the smallest and D the largest perpendicular tumor diameters. A contrast analysis using Bonferroni–Holm correction for multiplicity following a two-way analysis of variance (ANOVA) { XE “Anova” \f Abbreviation \t “analysis of variance” } with factor treatment and day (repeated) was performed on tumor volume changes from baseline, to compare globally and at each day, all treated groups to the control group and all treated groups with the combination group. A probability of less than 5% (*p* < 0.05) was considered significant. Statistical analysis was performed using EverStat6 software.

## 5. Conclusions

The addition of a TEAD inhibitor potentiates the effect of KRAS^G12C^i in a set of NSCLC cell lines. Mechanistically, combination treatment impacts significant cell cycle checkpoint events, leading to an accumulation of cells in G1 and a decrease in cells in G2/M, accompanied by the downregulation of SAC proteins. This effect is combination specific and does not occur with either single treatment.

## Figures and Tables

**Figure 1 pharmaceuticals-16-00553-f001:**
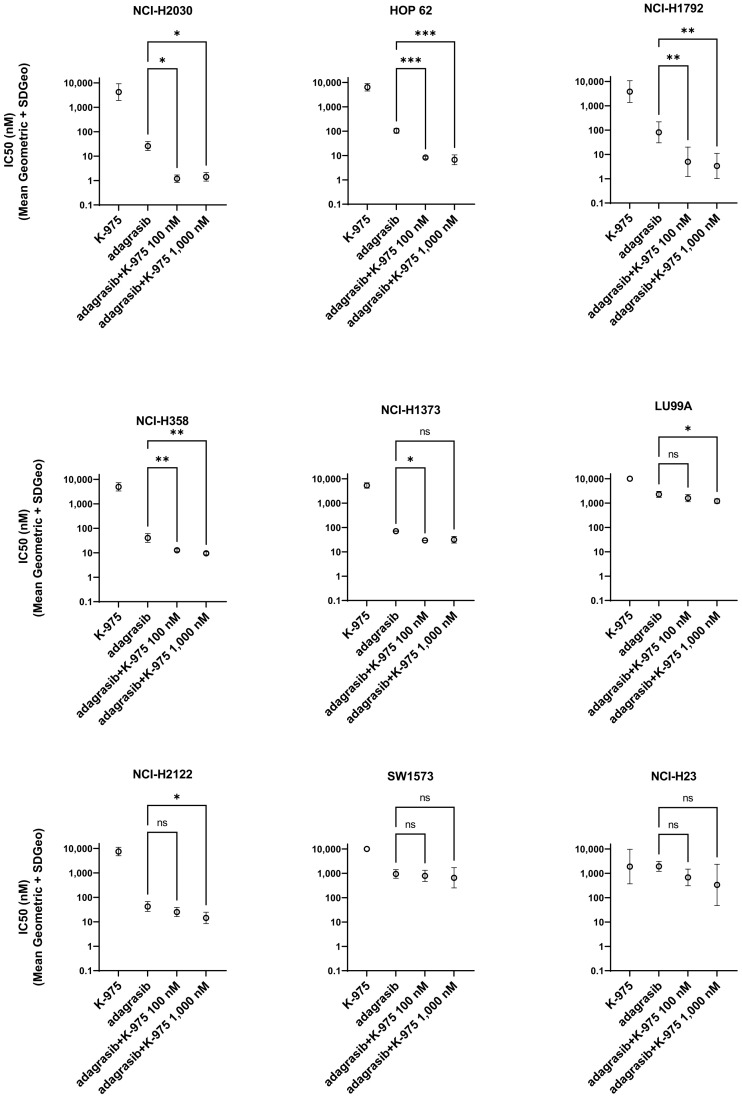
TEADi K-975 increases the response to KRAS^G12C^i adagrasib in several KRAS^G12C^-NSCLC cell lines. IC_50_ detection at 144 h after treatment for the single and combination treatments with adagrasib and K-975. The geometric mean and SD were calculated from a minimum of three biological replicates for each cell line, ns: non statistical significant, * *p*-value < 0.05, ** *p*-value < 0.001, *** *p*-value < 0.0005.

**Figure 2 pharmaceuticals-16-00553-f002:**
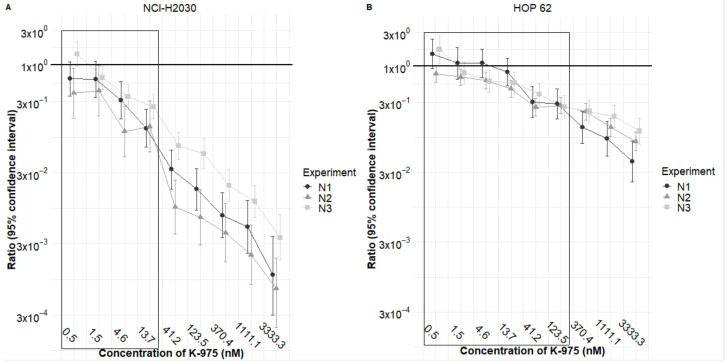
K-975 potentiates adagrasib in HOP 62 and NCI-H2030 cell lines. This graph shows the ratio values obtained for adagrasib either as a single agent or combined with increasing concentrations of the K-975 inhibitor. (**A**) Confirmatory study with the NCI-H2030 cell line shows the potentiation effect that is statistically evident for the four concentrations at which K-975 is entirely inactive as a single agent. (**B**) Confirmatory study with the HOP 62 cell line shows the potentiation effect that is statistically evident for six concentrations at which K-975 is entirely inactive as a single agent. The boxed values in panels A and B show the ratio of the potentiation effect of adagrasib when K-975 is used at doses where it is completely inactive as a single agent. Geometric means and SD were calculated for three independent experiments (N1, N2, N3).

**Figure 3 pharmaceuticals-16-00553-f003:**
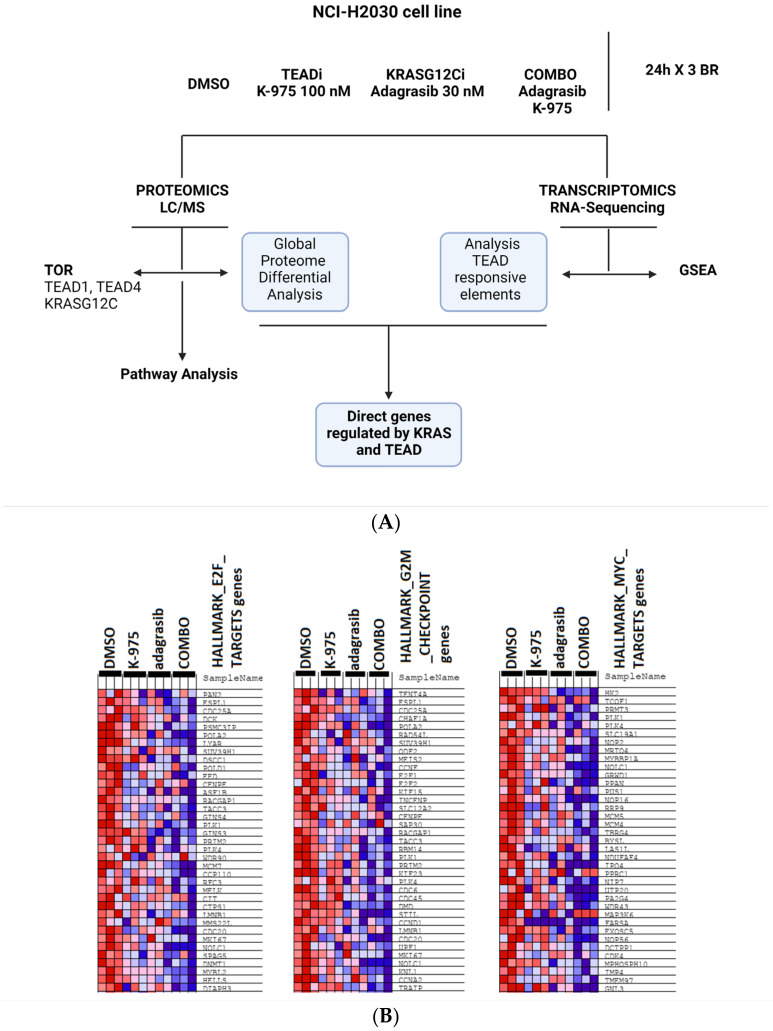
Transcriptomic and proteomic approaches show regulation in G1/S and G2/M transition checkpoints for the combined treatment of adagrasib and K-975. (**A**) Schematic representation of the conditions analyzed: DMSO, TEADi K-975 at 100 nM, KRAS^G12C^i adagrasib at 30 nM, COMBO (adagrasib and K-975). Three biological replicates (BR) of the four conditions described above were collected at 24 h. The treated cells were used to perform transcriptomic and proteomic analyses. TOR was observed in TEAD1, TEAD4, and KRAS^G12C^ pockets to see each compound’s occupancy in each condition analyzed. For the proteomic data, differential analysis was performed to see the affected pathways using Metacore. Gene set enrichment analysis (GSEA) and research of TEAD-responsive elements were performed from transcriptomic data. These data were finally crossed to see the direct target of our combination. (**B**) Representative GSEA of hallmark gene sets comparing all the treatments versus DMSO. The first three most modulated signatures are shown. Genes positively regulated are in red, while genes negatively regulated are in blue. For graphical matters, we cut the panels to show only the first set of 35–40 genes as representative figures FDR *q* value < 0.1. (**C**) Volcano plots showing proteomic data. Proteins modulated in the single or combination treatments with K-975 and/or adagrasib versus DMSO are shown. Downregulated proteins are represented as green dots and upregulated proteins as red dots. Highlighted in dark green and indicated by the arrows are PLK1, CDC20, DIAPH3, and BUB1B. FC > 2, *p* value < 0.01. (**D**) GO and pathway map in heatmap representation was performed from proteomic data and using Metacore. The first ten features are shown using FC > 2 and *p* value < 0.01.

**Figure 4 pharmaceuticals-16-00553-f004:**
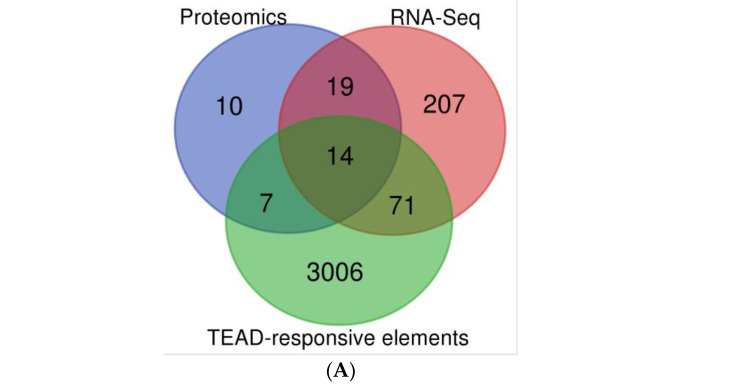
Gene selectively regulated in combination treatment with adagrasib and K-975. (**A**) Venn Diagram between RNA-Seq data, Proteomic data, and TEAD-Responsive elements from ChIP-Seq data. (**B**) Biomarker modulation at 24 h. Percent inhibition of proteins modulated in the conditions tested: DMSO, adagrasib at 30 nM, K-975 at 100 nM, and combination (COMBO; adagrasib and K-975). The data set is on three biological replicates.

**Figure 5 pharmaceuticals-16-00553-f005:**
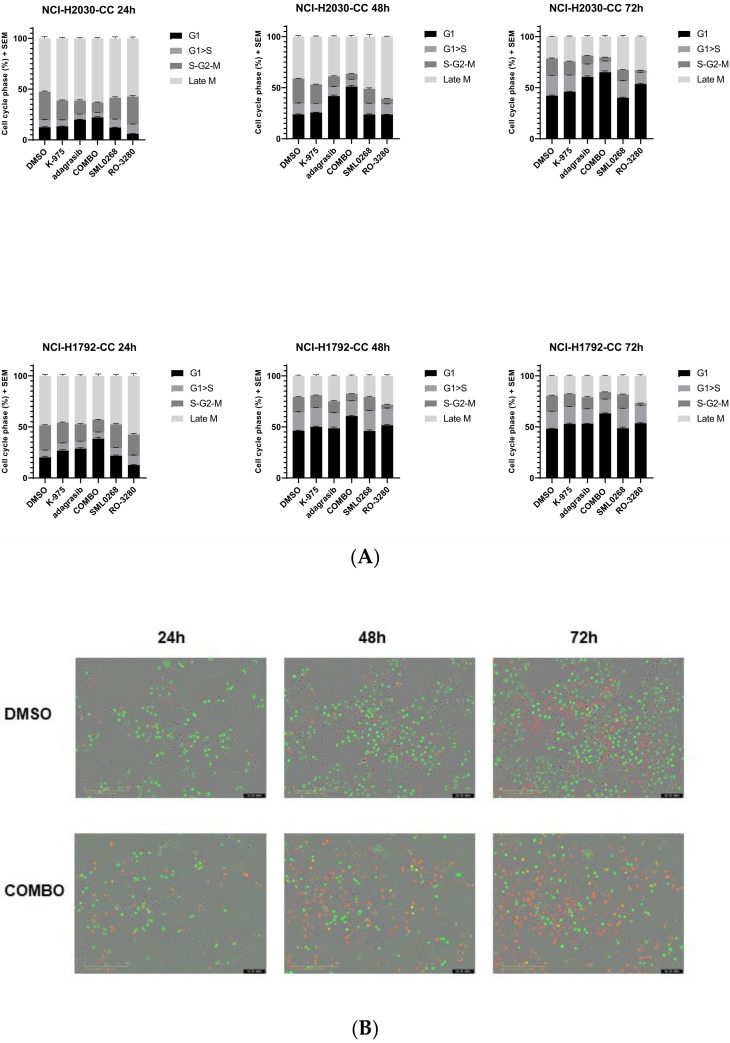
Cell cycle analysis and FACS detection show an arrest in the G0/G1 and G2/M phases. (**A**) Time course on cell cycle analysis. Histograms representing different phases of the cell cycle. The analysis was performed using NCI-H2030 (NCI-H2030-CC) and NCI-H1792 (NCI-H1792-CC) modified with lentivirus (Sartorius). The conditions analyzed were DMSO, K-975 at 100 nM, adagrasib at 30 nM, and COMBO (adagrasib and K-975 at the selected doses). Representative figures from at least two biological replicates (BR) of the different conditions were analyzed at 24, 48, and 72 h. Aurora B Kinase (SML0268) and PLK1 (RO-3280) inhibitors were used as control. Cell cycle phases analyzed were: G1, G1 > S (Transition), S-G2-M, and Late M. (**B**) Representative photos of time course on NCI-H2030-CC cell line. Conditions analyzed: DMSO and COMBO. The cell carpet is represented by cycling cells (green cells), while in the combination, the cell-confluency is rarer, and the cells are arrested in G1 (red cells). Details are described in Section 4. (**C**) Cell cycle analysis via FACS procedure at 48 h. Validation via propidium iodide of the DNA content. Conditions analyzed: DMSO, K-975 (100 nM), adagrasib (30 nM), COMBO (adagrasib and K-975). The image represents three biological replicates.

**Figure 6 pharmaceuticals-16-00553-f006:**
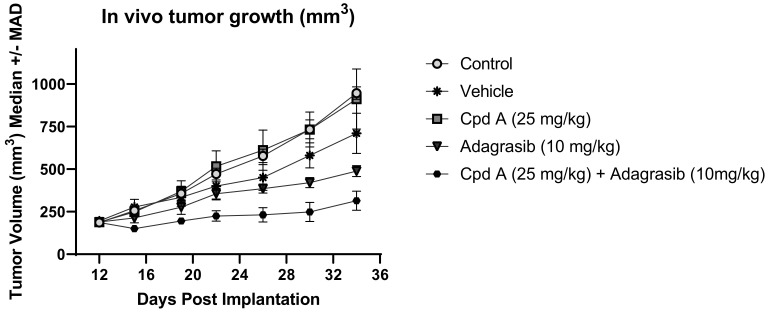
Sanofi internal Cpd A ameliorates adagrasib activity in tumor-bearing NCI-H358. Five conditions were analyzed: control, vehicle, Cpd A at 25 mg/kg, adagrasib at 10 mg/kg, and the combination of these last two compounds together. The treatments started 12 days after implantation, and samples were taken to compare the size of the tumors. Eight mice were used for each group.

**Figure 7 pharmaceuticals-16-00553-f007:**
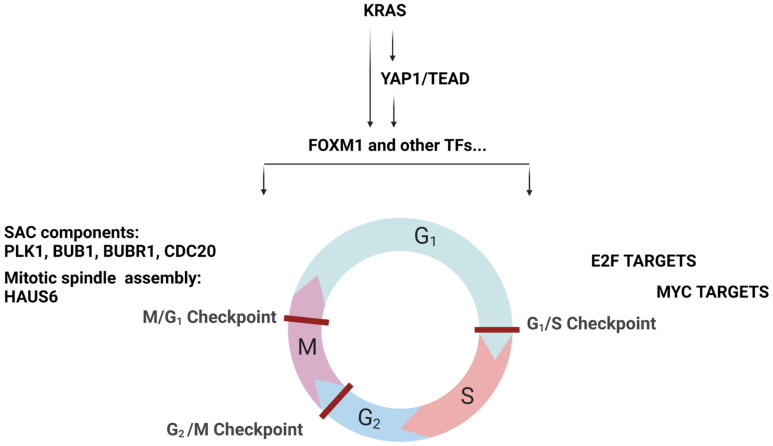
Potential molecular mechanisms explaining the crosstalk between KRAS and YAP1-TEAD. On the left-hand side of the panel, we propose that KRAS and YAP1-TEAD signaling converges essentially on the regulation of transcriptional activity of FOXM1, which in turn regulates some components of the SAC impacting the G2/M and M/G1 checkpoints. On the right-hand side of the panel, we propose that KRAS and YAP1-TEAD signaling may act on the E2F and MYC targets (via ATAD2 and HELLS). The modulation here may affect the G1/S checkpoint.

## Data Availability

The mass spectrometry proteomics data have been deposited to the ProteomeXchange Consortium (http://proteomecentral.proteomexchange.org (accessed on 15 February 2023)) with the dataset identifier PXD039726 and to the jPOSTrepository (https://repository.jpostdb.org (accessed on 15 February 2023)) with the dataset identifier JPST002012. The transcriptomic data from RNA-Seq have been deposited to GEO with the dataset identifier GSE224439. These data are scheduled to be publicly available on 9 February 2023 at: https://www.ncbi.nlm.nih.gov/geo/query/acc.cgi?acc=GSE224439 (accessed on 9 February 2023).

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
