# Peer review of "TEAD Inhibitors Sensitize KRASG12C Inhibitors via Dual Cell Cycle Arrest in KRASG12C-Mutant NSCLC"

_pharmaceuticals, 2023, doi:10.3390/ph16040553_

Round 1

Reviewer 1 Report

The paper titled "TEAD inhibitor K-975 enhances KRASG12C inhibitor adagrasib-mediated cancer cell proliferation inhibition via dual cell cycle arrest" describes the effectiveness of this combination treatment in causing cell cycle arrest in KRASG12C lung cancer cells. Using various in vitro assays coupled with transcriptomic and proteomics approaches, the authors derive the conclusion that the combination treatment causes a G1 arrest and a shortened G2/M phase. The overall experimental design has merit, but there is scope to improve the quality of the manuscript by properly describing some of the results and discussion. I would appreciate if the authors could respond to the following comments and suggestions:

 1. The abstract can be significantly improved. E.g. the sentence about YAP1 activation and MAPK inhibitors does not add value to the abstract. 

2. Line 87-89: It would be helpful if the authors provide data of 10 doses of adagrasib and two doses of K-975 used?

3. Line 95: What is meant by phenotypic combination study?

4. Why was H2030 selected for further phenotypic combination study? H358 and H1792 are more significantly inhibited by the combination treatment.

5. Please explain the rationale of using SML0268 and RO-3280 as it is not discussed properly.

6. Why was sotorasib also used in PTM scan assay? Do the authors intend to compare the two combination treatment? It is not clearly discussed.

7. For the in vivo study, the authors have used Cpd A instead of K-975. While the authors explain as to why it was used, it would be better for the reader to get a brief description in the introduction as well. Further, can the authors also provide survival data?

8. Apart from the two inhibitors mentioned in the title, the authors have used their alternatives in two of the important results. It should be clear to the reader by reading the abstract and introduction which compounds/inhibitors are used in the most relevant results. The authors are encouraged to accordingly improvise the title, abstract and introduction of the article.

9. Can the authors provide information on the type of cell death the cancer cells undergo during combination treatment? Is it apoptotic or necrotic cell death? 

10. Many inhibitor and combination treatments are also known to cause S-phase arrest. Is a G1 arrest coupled with G2/M declined better or worse than S-phase arrest? What are the authors thoughts?

11. Multiple studies have shown that synergistic effects of combination treatment could be a better treatment option for various type of cancer. How does the potentiation effect, as described here, compare to synergistic effect?

12. Please provide legend for supplementary figures as well.

Author Response

We would like to thank the reviewers for the time taken to review and evaluate our manuscript and for the pertinent comments and suggestions. We have addressed these comments and suggestions point by point below and have also made the requested changes to the manuscript. In addition, we have revised the English in the new version of the manuscript.

Answers to comments from Reviewer 1:

  1. The abstract can be significantly improved. E.g. the sentence about YAP1 activation and MAPK inhibitors does not add value to the abstract. 

Thank you for this comment, we have removed the sentence about YAP1 activation and MAPK inhibitors and have modified the abstract for better legibility.

  1. Line 87-89: It would be helpful if the authors provide data of 10 doses of adagrasib and two doses of K-975 used?

Thank you for this question. We are not sure, whether the requested information regards the exact doses used in the experiment or an example of the individual data points obtained from the study. We have therefore provided information on both points:

A – There is a 3.16x dilution factor between each of the ten doses used in this combination experiment ranging from the lowest dose of 0.3 nM up to the highest dose of 10,000 nM. We have added this information also to the materials and methods part.

B – The data for each of the concentrations tested for adagrasib and K-975 are reported in the pdf file attached to our replies. We have provided examples for the following cell lines NCI-H2030, NCI-H1792, and Lu99A. Please note that the compound names in these attached file are different from the manuscript: MIRATI stands for adagrasib and KIRIN stands for K-975.

  1. Line 95: What is meant by phenotypic combination study?

Thank you for this question. We acknowledge that the way we wrote the phrase can be confusing. We have therefore changed the name from “phenotypic combination study” to “cell growth inhibition study” in the new version of the manuscript. In fact, in this study, we wanted to see whether the cell growth inhibition (the phenotype) that we observed in the 96-well format for which we used 10 concentrations of the KRASG12Ci and two concentrations of the TEADi (shown in Figure 1) could be observed also in another format. We therefore performed the confirmatory study in a 384-well plate format which allowed us to cover a larger set of concentrations (18 concentrations of the KRASG12Ci and 14 concentrations of the TEADi) and to improve the statistical power of the potentiation results (as we show in Figure 2 A and B).

  1. Why was H2030 selected for further phenotypic combination study? H358 and H1792 are more significantly inhibited by the combination treatment.

This is a very important question, we actually observed that NCI-H2030 and NCI-H1792 cell lines presented the same strong sensitivity to the combination treatment since we have an improvement of the IC50 around 20-fold in both cell lines when adagrasib is combined with K-975 compared to adagrasib treatment alone. For NCI-H358 we observed a 4-fold improvement when the cells were treated with the combination. Therefore, we chose one of the most sensitive cell lines for the further mechanistic studies.

  1. Please explain the rationale of using SML0268 and RO-3280 as it is not discussed properly.

Thank you for this comment. This explanation is indeed missing in the previous version of the manuscript and we have now clarified this in the new version of the manuscript. Briefly, we decided to compare the effect on the cell cycle obtained with the combination of the KRASG12Ci and TEADi to the effect on the cell cycle obtained with an Aurora B kinase (SML0268) or a PLK1 (RO-3280) inhibitor, because each of these treatments (KRASG12Ci and TEADi combination, Aurora B inhibition, or PLK1 inhibition leads to the modulation of mitotic spindle checkpoint proteins. However, while Aurora B and PLK1 biomarkers are specifically downregulated by the KRASG12Ci and TEADi combination treatment, the combination treatment exerts an effect on the cell cycle that differs from the effect on the cell cycle obtained with either a PLK1 or an Aurora A/B inhibitor.

  1. Why was sotorasib also used in PTM scan assay? Do the authors intend to compare the two combination treatment? It is not clearly discussed.

Thank you for this question, and again it is true that we did not specify in the text why we used also this KRASG12C inhibitor. The use of sotorasib in the PTMScan assay was to confirm that a structurally different KRASG12C inhibitor would yield the same effect for our combination study, which  indeed was the case. The intention was not to specifically compare these two KRASG12C inhibitors.

  1. For the in vivo study, the authors have used Cpd A instead of K-975. While the authors explain as to why it was used, it would be better for the reader to get a brief description in the introduction as well. Further, can the authors also provide survival data?

Thank you for this remark, we have indeed included this information in the introduction of the new version of this manuscript. Regarding the in vivo study, we have not looked at survival as study was designed to look at compound efficacy on tumor xenograft growth. Mice were sacrificed once tumor volumes in the control group had reached the limit of what is ethically acceptable following the French law and the European Directive 2010/63/EU for the protection of animals used for scientific purposes and with approval of the ethics committee #21 (project number APAFIS#5644-2016061311593064.V1).

  1. Apart from the two inhibitors mentioned in the title, the authors have used their alternatives in two of the important results. It should be clear to the reader by reading the abstract and introduction which compounds/inhibitors are used in the most relevant results. The authors are encouraged to accordingly improvise the title, abstract and introduction of the article.

Thank you for this comment. We have indeed changed the title accordingly and have also reformulated the abstract and the introduction.

  1. Can the authors provide information on the type of cell death the cancer cells undergo during combination treatment? Is it apoptotic or necrotic cell death? 

Thank you for this very interesting question. We have preliminary data indicating that there was no apoptotic induction in the combination treatment and that the combination effect we observe leads to cell growth inhibition and not to cell death induction.

  1. Many inhibitor and combination treatments are also known to cause S-phase arrest. Is a G1 arrest coupled with G2/M declined better or worse than S-phase arrest? What are the authors thoughts?

This is a difficult question to answer and at this stage we cannot make the conclusion that a dual cell cycle arrest would be beneficial over a single cell cycle arrest in S-phase or M-phase phase. The main advantage from a drug discovery standpoint for this combination treatment lies in the fact that (i) the TEADi potentiates the effect of the KRASG12Ci and therefore renders the KRASG12Ci more potent or the cell lines more sensitive to this compound and that (ii) this effect should in theory be specific to tumor cells with KRASG12C mutations and is hence not expected to occur in normal (healthy) cells which a priori do not carry a KRASG12C mutation.

We can speculate, that the dual cell cycle arrest would potentially prevent rapid escape from treatment and delay acquired resistance as opposed to treatment with a KRASG12Ci alone. However, additional studies are needed to demonstrate this hypothesis and show whether a dual cell cycle inhibition could be favourable in this context.

  1. Multiple studies have shown that synergistic effects of combination treatment could be a better treatment option for various type of cancer. How does the potentiation effect, as described here, compare to synergistic effect?

We have used the term potentiation as recommended by our statistics department, as the term potentiation is used when one of the compounds tested is not active at all as a single agent. The term synergy is recommended in a context where both compounds exert activity as single agents. We can therefore not directly compare the potentiation effect to a synergistic effect because these two characterizations are not supposed to be used in the same context. In conclusion, potentiation can be seen as the equivalent to synergy in a setting where one of the two compounds is non active as a single agent.

  1. Please provide legend for supplementary figures as well.

Thank you for this comment, we have realised that the legends for the Supplementary Figures in the word file had shifted a bit, we have corrected this in the new version.

Reviewer 2 Report

The authors investigated the effects of the TEAD inhibitor K975 or CpdA in combination with KRASG12Ci in NSCLC cell lines and animals. This reviewer has the following suggestions to improve the MS:

1) TEAD1-4 expression needs to be confirmed in these cell lines. Since there is no difference in SW1573 and NCI-H23, this is especially important. Is there any correlation between TEAD expression levels and efficacy.

2) In Fig.3C,  it will be nice to validate the expression of the targets by RT-PCR and/or Western blotting analysis.

Author Response

We would like to thank the reviewers for the time taken to review and evaluate our manuscript and for the pertinent comments and suggestions. We have addressed these comments and suggestions point by point below and have also made the requested changes to the manuscript. In addition, we have revised the English in the new version of the manuscript.

Answers to comments from Reviewer 2:

1) TEAD1-4 expression needs to be confirmed in these cell lines. Since there is no difference in SW1573 and NCI-H23, this is especially important. Is there any correlation between TEAD expression levels and efficacy.

Thank you for this comment, which is indeed an important remark. We have generated TEAD1-4 expression data for the cell lines for which we had the complete RNA-Seq data available and have included this graph here below. In addition, we have looked at TEAD1-4 expression in the DepMap data set for all the cell lines used (in the file attached). Based on these analyses (our own RNA-Seq data and the DepMap) we observe, that all TEAD isoforms are expressed in the cell lines used for our studies and while there is some difference in the degree of expression at the RNA level, this difference does not allow to conclude on a correlation with the response to treatment. A larger cell panel would be required and ideally also the information on TEAD1-4 protein levels should be considered. Of note: K-975 and Sanofi cpd A bind all isoforms of TEAD.

2) In Fig.3C,  it will be nice to validate the expression of the targets by RT-PCR and/or Western blotting analysis.

Thank you for this remark, you will find the validation via Western Blot (Jess Technology) of some targets identified in the Figure .3C, in the Figure 4B, and in the supplementary Figure S2 A and B.

Round 2

Reviewer 1 Report

The authors have addressed all the questions and concerns raised.

Please check line no. 454, regarding dilution factor, for possible typo.

Overall, this version of manuscript can be accepted in its current form.

Reviewer 2 Report

Concerns have been largely addressed.